# Targeting the Gut Microbiota of Vertically HIV-Infected Children to Decrease Inflammation and Immunoactivation: A Pilot Clinical Trial

**DOI:** 10.3390/nu14050992

**Published:** 2022-02-26

**Authors:** Talía Sainz, Laura Diaz, David Rojo, María Isabel Clemente, Coral Barbas, María José Gosalbes, Nuria Jimenez-Hernandez, Luis Escosa, Sara Guillen, José Tomás Ramos, María Ángeles Muñoz-Fernández, María Luisa Navarro, María José Mellado, Sergio Serrano-Villar

**Affiliations:** 1Servicio de Pediatría, Hospital Universitario La Paz and IdiPAZ, 28046 Madrid, Spain; luisescosa1983@gmail.com (L.E.); mariajose.mellado@salud.madrid.org (M.J.M.); 2Red de Investigación CoRISpe Integrada en la Red en Infectología Pediátrica (RITIP), 28046 Madrid, Spain; sguillen@salud.madrid.org (S.G.); josetora@ucm.es (J.T.R.); mmunoz.hgugm@gmail.com (M.Á.M.-F.); marisa.navarro.gomez@gmail.com (M.L.N.); 3Área de Enfermedades Infecciosas del Centro de Investigación Biomédica en Red del Instituto de Salud Carlos III (CIBERINFEC), Instituto de Salud Carlos III, 28029 Madrid, Spain; serranovillar@gmail.com; 4Plataforma de Citometría (Laura Diaz), Instituto de Investigación Sanitaria Gregorio Marañón (IsSGM), Hospital Universitario Gregorio Marañon, 28009 Madrid, Spain; lauradiaz@gmail.com (L.D.); maribel.clemente@iisgm.com (M.I.C.); 5Unidad de Cultivos Celulares (María Isabel Clemente), Instituto de Investigación Sanitaria Gregorio Marañón (IsSGM), Hospital Universitario Gregorio Marañon, 28009 Madrid, Spain; 6Laboratorio de Inmunobiología (Maria Angeles Muñoz-Fernandez), Instituto de Investigación Sanitaria Gregorio Marañón (IsSGM), Hospital Universitario Gregorio Marañon, 28009 Madrid, Spain; 7Centre for Metabolomics and Bioanalysis (CEMBIO), Department of Chemistry and Biochemistry, Facultad de Farmacia, Universidad San Pablo-CEUUniversities, Boadilla Monte, 28668 Madrid, Spain; davidrb87@gmail.com (D.R.); cbarbas@ceu.es (C.B.); 8Área Genómica y Salud, Fundación para el Fomento de la Investigación Sanitaria y Biomédica (FISABIO), 46020 Valencia, Spain; maria.jose.gosalbes@uv.es (M.J.G.); nuriajimenez@gmail.com (N.J.-H.); 9CIBER en Epidemiología y Salud Pública (CIBEREsp), Instituto de Salud Carlos III, 28029 Madrid, Spain; 10Servicio de Pediatría, Hospital de Getafe, 28901 Madrid, Spain; 11Servicio de Pediatría, Hospital Clinico San Carlos e Instituto de Investigación Sanitaria del Hospital Clínico San Carlos (IdISSC), 28040 Madrid, Spain; 12Departamento de Salud Pública y Materno-Infantil, Universidad Complutense de Madrid (UCM), 28040 Madrid, Spain; 13Servicio de Pediatría, Hospital General Universitario Gregorio Marañón, 28009 Madrid, Spain; 14Servicio de Enfermedades Infecciosas, Hospital Universitario Ramón y Cajal, and IRYCIS, 28034 Madrid, Spain

**Keywords:** HIV, inflammation, immunoactivation, microbiota, children

## Abstract

Aims: Children with HIV exhibit chronic inflammation and immune dysfunction despite antiretroviral therapy (ART). Strategies targeting persistent inflammation are needed to improve health in people living with HIV. The gut microbiota likely interacts with the immune system, but the clinical implications of modulating the dysbiosis by nutritional supplementation are unclear. Methods: Pilot, double-blind, randomized placebo-controlled trial in which 24 HIV-infected on ART were randomized to supplementation with a daily mixture of symbiotics, omega-3/6 fatty acids and amino acids, or placebo four weeks, in combination with ART. We analyzed inflammatory markers and T-cell activation changes and their correlations with shifts in fecal microbiota. Results: Twenty-four HIV-infected children were recruited and randomized to receive a symbiotic nutritional supplement or placebo. Mean age was 12 ± 3.9 years, 62.5% were female. All were on ART and had HIV RNA < 50/mL. We did not detect changes in inflammatory (IL-6, IL-7, IP-10), microbial translocation (sCD14), mucosal integrity markers (IFABP, zonulin) or the kynurenine to tryptophan ratio, or changes in markers of the adaptive immune response in relation to the intervention. However, we found correlations between several key bacteria and the assessed inflammatory and immunological parameters, supporting a role of the microbiota in immune modulation in children with HIV. Conclusions: In this exploratory study, a four-week nutritional supplementation had no significant effects in terms of decreasing inflammation, microbial translocation, or T-cell activation in HIV-infected children. However, the correlations found support the interaction between gut microbiota and the immune system.

## 1. Introduction

It is well known that HIV-infected children present persistent immune dysfunction despite early antiretroviral treatment (ART) [1,2,3,4]. Among adult populations, recent studies have pointed out the potential influence of the microbiome [5,6,7,8]. Different interventions aimed at mitigating the complex HIV-related gut-associated lymphoid tissue defects have been attempted to reduce the long-term consequences of chronic inflammation, mostly among adults [9,10]. Still, the effects of long-term immune abnormalities in the context of vertical infection are barely understood [1,4].

During acute HIV infection, massive depletion of lymphocyte populations occurs in the gut, only partially restored by ART [5,8,11,12]. In the last years, several studies have underlined the potential role of not only subsequent microbial translocation but also gut microbiota-immune system interactions in the persistent immune dysfunction associated with chronic HIV infection [5,11]. Although there is some controversy regarding the pattern of dysbiosis among people living with HIV, most studies addressing gut microbiota in this population describe significant changes in comparison to uninfected controls [8,9,13,14], some of them potentially promoting inflammatory pathways [6,7,11]. While first studies describing gut microbiota composition pointed towards an altered ecosystem also among children, whose microbiome settles already in the presence of an altered immune system, results are controversial in terms of the extent of gut dysbiosis [15,16]. The persistence of thymic function in children may hypothetically play a role for gut-associated-lymphoid tissue (GALT) immune reconstitution despite HIV infection [17], leading to differences in terms of dysbiosis when compared to the infection acquired later in life. Furthermore, childhood is known to be key in terms of acquisition of the human microbiome, which seems to establish over the first years of life [18]. Not surprisingly, recent evidence has raised the question of whether childhood might be the optimal window for intervention targeting microbial communities, potentially impacting the bacteria-immune system interplay, influencing different metabolic and inflammatory conditions. Nevertheless, few attempts targeting the microbiome have included children [19], and none to our knowledge has included symbiotics.

Strategies aimed at reducing chronic inflammation are also needed for children living with HIV. On the basis that the gut microbiota-immune system interactions likely influence immune dysfunction during HIV infection, we aimed to assess the potential impact on chronic inflammation, microbial translocation, and immunoactivation of a nutritional supplementation aimed at modulating the dysbiosis of perinatally HIV-infected children.

## 2. Methods

### 2.1. Study Design, Participants, Setting and Eligibility

We conducted a pilot, double-blind, randomized, placebo-controlled study. Perinatally HIV-infected children and adolescents aged 6 to 18 years, on stable ART and virologically suppressed for at least 6 months, with CD4+ T-cell counts ≥350 cells/µL, were enrolled. Patients were under follow-up at the Outpatient clinics of four Hospitals in Madrid, Spain (Hospital La Paz, Hospital Clínico San Carlos, Hospital Gregorio Marañón y Hospital de Getafe), between October 2013 and November 2014. Exclusion criteria included: use of systemic antibiotics during the previous three months; concomitant medications; and any acute or chronic condition other than chronic HIV infection. Patients co-infected with hepatitis B or C viruses were excluded. At inclusion, cases were randomized to receive either a nutritional supplement designed ad hoc (PMT25341) or the placebo, daily for four weeks. Peripheral blood samples and fecal samples were collected at baseline and after the four weeks intervention (±7 days).

The study protocol was approved by the Independent Ethics Committees at all participating institutions (approval number 173/13) and all parents/tutors and participants above 12 years of age provided written informed consent/assent. The protocol was conformed to the principles of the Declaration of Helsinki and the Good Clinical Practice Guidelines.

### 2.2. Nutritional Intervention

A specifically designed powder formulation (PMT25341) [20] containing *Saccharomyces boulardi*, fructooligosaccharides, galactooligosaccharides, eicosapentaenoic acid, docosahexaenoic acid, linoleic acid, glutamine, arginine, AM3, and vitamin D was administered (20 g per day). The placebo was skimmed milk powder. Both products were manufactured by Nutricion Médica, S.L. and distributed to the centers in indistinguishable individual packages that were delivered to the patients to be administered daily mixed with food or beverages for 4 weeks. The study participants were randomized by a computer-generated randomized number system in blocks. All participating clinicians and researchers, as well as the children and their families, were blind to the assigned patient group.

### 2.3. Laboratory Assessments

Fasting blood samples were drawn for real-time plasma HIV-1 viral load measurements and immunological studies. HIV viral load (VL) was quantified using the Cobas TaqMan HIV-1 assay (Roche Diagnostics Systems, Inc., Branchburg, NJ, USA) with a detection limit of 50 copies/mm^3^. Immunophenotyping was performed from fresh samples: CD4, CD8 T-cell counts, and T-cell subpopulations were measured with standard flow cytometric methods. T regulatory cells were characterized by the expression of CD25⁺CD127^−^. T-cell activation was characterized by HLA-DR⁺ and CD38⁺ expression, senescence by CD28^−^CD57⁺ expression, and exhaustion by PD-1. Stained cells were run on a Gallios flow cytometer (Beckman Coulter, Inc., Münster, Germany), and Kaluza software was used for data analysis (Beckman Coulter, Inc., Münster, Germany). A sample of every participant was sent to the Pediatric H.I.V. BioBank-HGUGM, and plasma was stored at −80° using standard procedures for the determination of inflammatory markers.

IL-6 and IP-10 were determined using multiplex immunoassays by Invitrogen™ ProcartaPlex™ (ThermoFisher, Waltham, MA, USA) based on the principles of a sandwich ELISA with the use of Luminex^®^ xMAP^®^. Additional ELISAs were performed for IL-7 (R&D IL-7 Quantikine HS ELISA Kit, R&D Systems, Minneapolis, MN, USA); intestinal fatty acid binding protein (IFABP), as a tissue-specific injury marker (R&D IFABP ELISA Kit, R&D Systems, Minneapolis, MN, USA), and soluble CD14 (sCD14) (R&D sCD14 Quantikine ELISA Kit DC140, R.D.D.) as a marker of monocyte activation/microbial translocation. Zonuline (Abyntek Biopharma S.L. ELISA Kit, Derio, Vizcaya, Spain) was measured as a marker of mucosal barrier integrity.

The plasma kynurenine to tryptophan ratio (KT ratio) was determined by mass spectrometry, as a measure of the activity of the indoleamine 2,3-dioxygenase (IDO) enzyme. Each analysis was achieved using a liquid chromatography system consisting of a degasser, binary pump, and autosampler (1290 Infinity, Agilent Technologies, Santa Clara, CA, USA) coupled to a triple quadrupole mass spectrometer (6460, AgilentTechnologies).

### 2.4. Gut Microbiota Analysis

Fecal samples were collected in sterile tubes with RNAlater (Life Technologies, Austin, TX, USA) and frozen at −80 °C. Total DNA was extracted from bacterial pellet as previously described [20]. For each sample, the V3-V4 region of the 16S rRNA gene were amplified from total DNA, and amplicon libraries were constructed following Illumina instructions. The alpha diversity was determined at the Operational Taxonomic Units (OTUs) using vegan library in R package. To analyse beta diversity, we applied Principal Coordinates analysis (PCoA) based on weighted and unweighted UniFrac distances in R package.

### 2.5. Statistical Analysis

Between-group comparisons of continuous variables were analyzed using the Mann-Whitney U test, and to evaluate differences in numerical outcomes between time-points we used the Wilcoxon signed-rank matched-pairs test. A cross-sectional analysis was performed to address baseline correlations between microbial composition/diversity and immunological/inflammatory variables. The Spearman correlation was used to assess correlations between biomarkers. Fold changes between baseline and week four measurements were calculated to analyze correlations between inflammatory and T-cell biomarkers and microbiome-associated biomarkers. Due to the reduced sample size, the exploratory design and to improve interpretability [21], we decided to select a few biomarkers from the microbiota dataset without adjusting for multiple comparisons rather than computing the correlations for the whole microbiota dataset. Statistical analysis was performed using Stata v17.0 (StataCorpL.P.P, College Station, TX, USA) and R package for the correlation analyses (library *corrplot*). Prism v.7.0, GraphPad, Inc., La Jolla, CA, USA), was used to create figures.

## 3. Results

### 3.1. Characteristics of the Study Population

Twenty-four vertically HIV-infected children and adolescents were recruited and randomized. The median age was 12 ± 3.9 years, 62.5% were female. All were on ART and virologically suppressed. Two patients, one in the intervention and one in the placebo arm did not complete the follow-up and were withdrawn from the study. Despite randomization, there was a significant difference at baseline between CD4 counts, which were higher in the intervention group at baseline. CD4 nadir was also significantly higher in the intervention group, with a non-significant difference in the CD4/CD8 ratio. The main characteristics of study participants are shown in Table 1.

### 3.2. Analysis of Inflammatory Biomarkers

At baseline, levels of inflammatory biomarkers were comparable between groups, with the exception of I-FABP levels, which were higher in the intervention group. Figure 1 represents the levels of soluble biomarkers at baseline and after intervention in each group. No within or between-group differences were appreciated in IL-6 or IL-7, both considered important cytokines for inflammation and immunoregulation, over the study period. The K/T ratio, a marker of indoleamine-2,3-dioxygenase-1 (IDO-1) induction involved in the regulation of mucosal immunity [22] remained stable over time. The baseline difference in IFABP between groups, a marker increasing after the loss of enterocyte barrier integrity and a predictor of clinical progression in treated HIV [23], was not observed at the end of the study, but changes over time within the groups were non-significant. We neither observed differences within nor between groups in sCD14.

### 3.3. Analysis of Immune Cell Phenotypes

As previously mentioned, CD4 T cell counts were significantly higher in the intervention group at baseline. No significant differences were identified when comparing changes over time according to the intervention group for CD4+, CD8+ T cell counts, or the CD4/CD8 ratio (Figure 2), and the difference in CD4 counts between intervention arms disappeared during the follow-up. We did not detect differences in the percentage of CD4+ or CD8+ T cells expressing markers of activation (HLADR+ CD38+), senescence (CD28-CD57+), and exhaustion (PD-1) (Figure 2), or in the naïve/memory subsets, with similar findings for the frequency of B or NK cells (data not shown).

### 3.4. Links between Shifts in the Microbiota and Changes in Systemic Markers

We previously published the comparison of microbiota composition in children with HIV vs. their uninfected peers, along with the effects of the nutritional intervention [20]. In summary, while we did not appreciate clear differences in alpha diversity at baseline or after the intervention when comparing HIV-infected vs uninfected children, or before and after nutritional supplementation. Differences in beta diversity at baseline between HIV-infected and uninfected controls suggested an effect of HIV on overall microbiota structure. The fact that the differences disappeared over the study period suggested mild attenuation of the compositional changes in the microbiota associated with HIV infection, yet differences were non-significant between intervention arms. Children with HIV showed enrichment for *Prevotella*, and *Akkermansia,* and depletion of *Faecalibacterium* and *Lachnospira,* among others.

We then asked whether longitudinal variations in the microbiota, regardless of the effect of the nutritional intervention, could affect the immune parameters measured in plasma and PBMC. Thus, we performed targeted correlation analyses with fold changes of key microbiota markers of HIV (alpha diversity and 13 selected genus based on our previous findings and the background literature) and the immunological parameters measured in plasma and PBMCs. We found several significant associations between variations in these parameters (Figure 3). Direct correlations included changes in alpha diversity and CD4/CD8 ratio (Figure 4); *Faecalibacterium* with IL-7, % naïve CD4+ and CD8+ T cells; *Lachnospira* with CD4+ T cells and %memory CD8+ T cells (Figure 5). Inverse correlations included alpha diversity with naïve CD4+ and CD8+ T cells (Figure 4); *Coprobacillus* with IFABP; *Lachnospira* with IL-6; *Eubacterium* and *Dorea* with CD8+ T cell counts and %memory T cells; and *Ruminococcus* with NK cells (Figure 5).

## 4. Discussion

In this pilot, controlled trial evaluating a short nutritional intervention based on the administration of a mixture containing prebiotics, probiotics, essential amino acids, and oligonutrients in children with HIV, we did not detect effects on microbial translocation, inflammation, and immunoactivation. However, our data support an implication of several key bacteria in relevant immune parameters, supporting a role of the microbiota in immune modulation in children living with HIV.

Several studies have evaluated dietary supplementation in adults living with HIV with various nutritional products, such as prebiotics and probiotics, among others [24,25,26]. However, the two large, controlled trials assessing the effects of probiotics or symbiotics on adults with HIV on ART failed to detect differences in any of the outcomes assessed, including the microbiota composition [24,25]. To the best of our knowledge, repeated fecal microbiota transplantation is the only intervention that so far demonstrated ability to induce long-lasting changes on the gut microbiota in adults living with HIV [27]. The lack of an effect on microbiota composition after prebiotic or probiotic supplementation in larger and longer studies in HIV-infected adults, compared to evidence of an effect in a smaller and shorter study in children, was encouraging for pediatric HIV research. This finding was in line with the hypothesis that the microbiota is less resilient in children than in adults, and a better scenario to assess strategies aiming to improve immune dysfunction by shaping the microbiota [20].

Most studies assessing dietary interventions in children with HIV have been designed to assess the impact on malnourishment and diarrhea [28,29]. The information regarding the effects of dietary interventions to target inflammation and persistent immune defects in children with HIV is very scarce. In a controlled study in 21 children with HIV in Iran, a 6-day supplementation with *Lactobacillus plantarum* resulted in decreases of LPS level with no changes on T cell counts, or fecal sIgA [30]. In a study in 60 children with HIV with and without HIV, both naïve and on ART, supplementation with fermented milk for 8 weeks resulted in changes in several T cell subsets, although in absence of a placebo group it is unclear whether these findings might be attributable to the nutritional intervention [19].

While we failed to detect an effect of a nutritional intervention on the immune parameters, our study was useful to evaluate the associations between longitudinal variations of key microbiota components and inflammatory and immunological parameters. To avoid a large number of comparisons in an unpowered study, we selected bacteria that were differentially abundant in our previous analysis in this cohort of children with HIV (depleted: *Faecalibacterium, Lachnospira, Coprococcus, Dorea, Lactococcus, Anaerostipes*; enriched: *Coprobacillus*) [20], along with other key relevant taxa (*Akkermansia, Bacteroides, Bifidobacterium, Butyricicoccus, Eubacterium, Prevotella, Roseburia,* and *Ruminococcus* highlighted as relevant in people living with HIV [8,10,31,32,33]. The findings summarized in Figure 3, Figure 4 and Figure 5 deserve further comment. First, increasing alpha diversity correlated with amelioration of the CD4/CD8 ratio. This finding is novel, and because a lower CD4/CD8 ratio is predictive of higher risk of clinical progression in adults with HIV [34], and increased T cell activation and senescence in children with HIV [4], this finding suggests that increasing alpha diversity could be interpreted as a sign of microbiota restoration in children with HIV, a question that is still debated in adults with HIV [32]. Second, *Lachnospira* increases correlated with IL-6 decreases, a robust predictor of mortality in HIV-infected adults [35], increases of CD4+ T cell counts, and a shift towards the memory compartment of CD8+ T cell maturational subsets. Interestingly, variations of *Ruminococcus* and *Faecalibacterium* abundance, belonging to the Ruminococcaceae family, and *Blautia, Dorea, Eubacterium, and Lachnospira*, belonging to the Lachnospiraceae family, were associated with changes in the immunological parameters measured in plasma and PBMC. These families comprise the major butyrate producers, which are thought to crucially influence human physiology [36]. Butyrate is considered a fundamental energy source of the colon epithelial cells since 70% of their energy is obtained from the oxidation of this acid [37]. Butyrate is implicated in maintaining gut homeostasis by promoting immunotolerance to commensal bacteria via down-regulation of lipopolysaccharide-induced pro-inflammatory mediators [38]. Hence, our data further support the butyrate synthesis pathway as a relevant mechanism mediating host-microbiota interactions in children with HIV. Last, decreases of *Lactobacillus*, which was a genus significantly increased in these children with HIV compared to their uninfected siblings [20], correlated with IFABP decreases, a biomarker of enterocyte integrity, which has been shown to independently predict mortality in adults with HIV [23]. Whether the nutritional intervention failed to improve these immunological predictors of disease progression, these findings support a link between changes in the microbiota and predictors of disease progression in children with HIV and encourage further research in this field.

The fact that, for safety reasons, a basal CD4 T cell count above 350 cell/mm was required for inclusion, excluded immunological non-responders from participation, which may be the population that hypothetically benefits most from an intervention addressing immune dysfunction. In fact, CD4+ T cell counts were significantly increased at baseline in the intervention group, and this could have limited the impact of the intervention. As immune cell subsets typically evolve slowly under stable conditions, we interpret that detecting changes at this level was difficult given the short course of the intervention, coupled with the good immunological situation of the study participants.

The main strengths of our study include (i) the inclusion of a placebo arm that allowed a better interpretation of the results, (ii) the selection of uninfected siblings as controls, which diminished the potential confounding effect of genetic and lifestyle factors, (iii) the novelty and relevance for the field of immunonutrition of assessing interventions in childhood, in which microbial communities have not yet achieved a stable configuration [39], (iv) the comprehensive assessment of multiple biomarkers of inflammation and adaptive immunity. The main limitations are inherent to exploratory studies and include the small sample size and the short duration of the intervention, which might have limited our ability to detect effects.

In conclusion, in this pilot study exploring the effect of a short dietary intervention to improve immune dysfunction among vertically HIV-infected children we did not detect effects on markers of inflammation and the adaptive immune response. However, the correlations found between several key bacteria and the assessed immune parameters support a role of the microbiota in the regulation of chronic inflammation and immune modulation in children with HIV. Strategies aimed at downregulating inflammation secondary to HIV infection are needed to improve the health of perinatally HIV-infected children, with life-long exposure to the infection. Our findings could help in the design of future studies evaluating approaches to target persistent immune defects via modulation of the gut microbiota.

## Figures and Tables

**Figure 1 nutrients-14-00992-f001:**
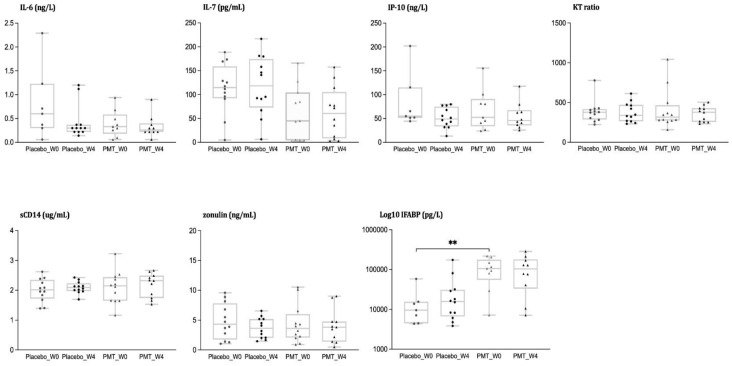
Inflammation, microbial translocation, and intestinal permeability markers. Determinations of IL-6, IL-7, IP-10, K.T.T. ratio, sCD14, zonulin, and IFABP at baseline and week 4 in HIV-infected children according to the intervention group (placebo vs. PMT25341). ** Statistical significance < 0.01.

**Figure 2 nutrients-14-00992-f002:**
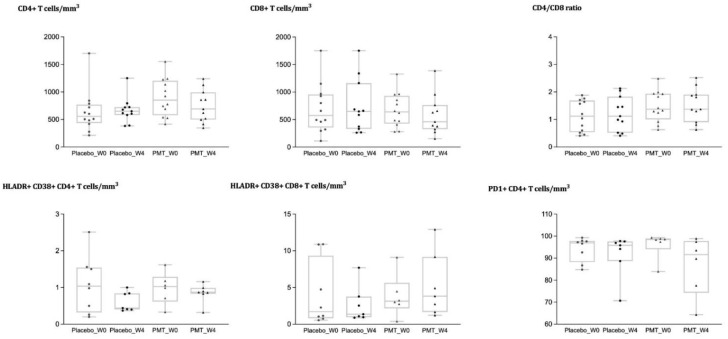
Changes in immune cell phenotypes in blood according to the intervention group (placebo vs. PMT25341). CD4, CD8, CD4/CD8 ratio, activation markers on CD4 and CD8 T cells, and exhaustion in CD4 T cells, at baseline and week 4 in HIV-infected children according to the intervention group (placebo vs. PMT25341).

**Figure 3 nutrients-14-00992-f003:**
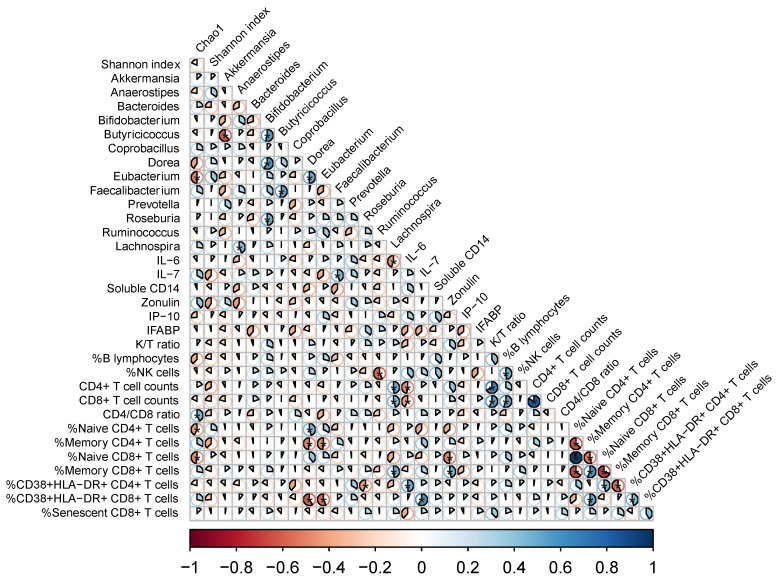
Heat map of targeted correlations between immunological parameters, and the microbiota (alpha diversity and 13 selected taxa). We considered the fold changes from the baseline for each parameter. The pie charts represent the magnitude of each individual Spearman Rho correlation coefficient in a color gradient from red (Rho − 1) to blue (Rho + 1). Correlations with a *p* < 0.05 are marked with an asterisk.

**Figure 4 nutrients-14-00992-f004:**
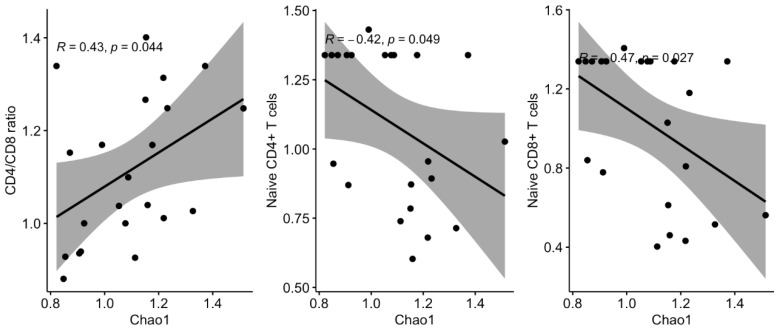
Scatter plots and regression lines for the significant correlations between the immunological parameters and the microbiota (alpha diversity). We considered the fold changes from baseline for CD4/CD8 ratio, naive CD4 T cells, naïve CD8 T cells and Chao1 index.

**Figure 5 nutrients-14-00992-f005:**
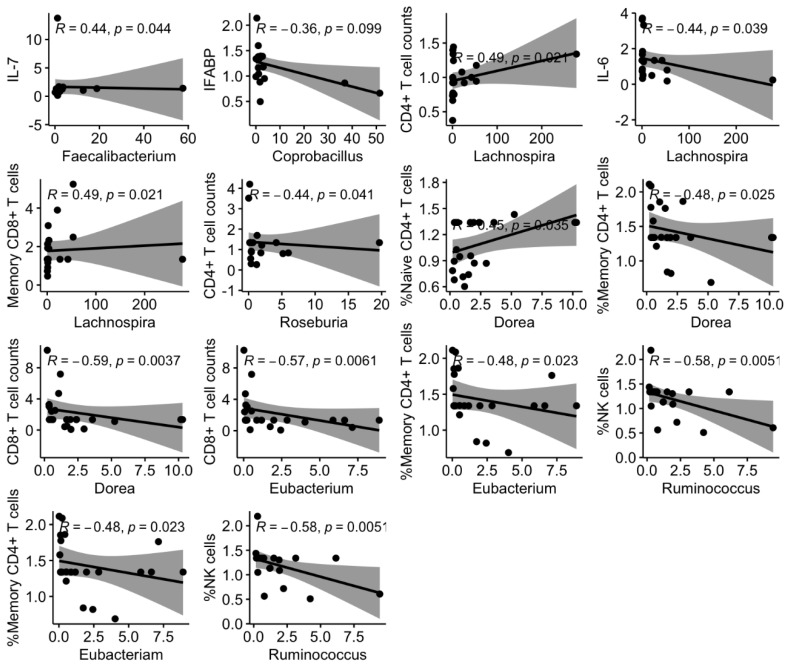
Scatter plots and regression lines for the significant correlations between the immunological parameters, and the microbiota (13 selected taxa). We considered the fold changes from the baseline for each parameter.

**Table 1 nutrients-14-00992-t001:** Main characteristics of the study participants at baseline according to the intervention group.

	Placebo *N* = 12	Nutritional Intervention *N* = 12	*p*
Female (N,%)	7 (58.3)	8 (66.7)	1.000
Age (years), mean (SD)	13.8 (3.6)	10 (3.4)	0.064
Caucasian (N, %)	6 (50)	7 (58.3)	1.000
CD4 count (cells/mm^3^)	556 (453–754)	852 (617–1182)	0.0496
CD4/CD8 ratio	1.1 (0.56–1.67)	1.4 (1.09–1.94)	0.106
CD4 Nadir (cells/mm^3^)	333 (169–382)	519 (384–979)	0.006
PI based ART (N, %)	8 (66.7)	9 (75)	0.136

All values are expressed in median (IQR) except otherwise specified. ART: antiretroviral treatment. PI: protease inhibitor.

## Data Availability

The data presented in this study are available upon reasonable request to the corresponding author.

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
