# Peer review of "Targeting the Gut Microbiota of Vertically HIV-Infected Children to Decrease Inflammation and Immunoactivation: A Pilot Clinical Trial"

_nutrients, 2022, doi:10.3390/nu14050992_

Round 1
Reviewer 1 Report
HIV is a model disease correlated with the immune system and many investigations were performed on the analysis of gut microbiota of patients infected with HIV. In this research, variations of key microbiota components and inflammatory and immunological parameters are investigated and correlations are searched.
Research is well done, I have only some observation:
In the beginning of the paper, the abbreviation ART is used, without any information on the meaning. The abbreviation ART should be explained in text, too, not only in Abstract. The same observation for PMT25341. Other abbreviations, as GALT, are not at all explained in text.
The experimental analyses are well described, but I consider that this chapter should be improved, because the authors don’t give an explanation on why and how they choose the number of 24 for the patients involved in the experiment. Why 24 is considered the adequate number of patients needed to answer the research question here?
Lack of indication in Figure 5 on what is indicated on abscissa.
Author Response
HIV is a model disease correlated with the immune system and many investigations were performed on the analysis of gut microbiota of patients infected with HIV. In this research, variations of key microbiota components and inflammatory and immunological parameters are investigated and correlations are searched.
Research is well done, I have only some observations:
In the beginning of the paper, the abbreviation ART is used, without any information on the meaning. The abbreviation ART should be explained in text, too, not only in Abstract. The same observation for PMT25341. Other abbreviations, as GALT, are not at all explained in text.
We appreciate the remark. The manuscript has been revised and all abbreviations are explained both in the abstract and the main text.
The experimental analyses are well described, but I consider that this chapter should be improved, because the authors don’t give an explanation on why and how they choose the number of 24 for the patients involved in the experiment. Why 24 is considered the adequate number of patients needed to answer the research question here?
This is a very important point. As mentioned in the introduction, data are very scarce regarding interventions with symbiotics in the unique population of vertically HIV-infected children. This study was design as an exploratory trial, and based on previous studies describing significant changes in the microbiota of HIV-infected adults, we stablished a sample size of 30 patients for this pilot clinical trial. Unfortunately, the recruitment was very challenging and only 24 children met the inclusion criteria. This may have limited our ability to find significant differences between the intervention arms, and is the main limitation here, as stated in the discussion and conclusion.
Lack of indication in Figure 5 on what is indicated on abscissa
We sincerely appreciate the remark. The figure has been corrected, thank you.
Reviewer 2 Report
Thank you for allowing me to review this manuscript. This manuscript entitled "Targeting The Gut Microbiota of Vertically HIV-infected Children to Decrease Inflammation and Immunoactivation: a Pilot Clinical Trial", aims to evaluate the potential impact on chronic inflammation, microbial translocation and immunoactivation of a nutritional supplement intended to modulate dysbiosis in children with perinatal HIV infection
It is an interesting and highly relevant article today, although it has several limitations that make it suitable for publication in this journal. These limitations are detailed below:
- In the introduction, it would be important to justify in more detail the importance and timeliness of the topic of study.
- In the material and methods section it is described in detail. It reflects exactly how the sample was collected, the inclusion and exclusion criteria, the variables and the software used. All this can be considered as a strength of the present manuscript. However, as a signal and weakness, we can highlight that different important ethical considerations are not reflected. It would be necessary to indicate if the participation of the patients was voluntary or if they were offered a document with the information and informed consent. Furthermore, it would be important to point out whether the sample is representative.
The results are presented in a clear and orderly manner. Also, an interpretation of them is reflected. However, Figure 1 (line 187) and Figure 2 (line 199) display poorly. Perhaps you could improve this point.
A section that gives the article quality, being a strength of it, is the discussion. In the manuscript, the results are discussed in an orderly manner, with necessary citations and rigor.
In relation to the conclusions, they are clear and precise. However, we recommend indicating more precisely the implication in clinical practice of the same. Also, it would be interesting to state the lines of the future that the authors consider.
Author Response
Thank you for the positive appreciation.
- In the introduction, it would be important to justify in more detail the importance and timeliness of the topic of study.
We appreciate the remark and have tried to emphasize the need for interventions addressing inflammation in people living with HIV, which as suggested justify the need of studies such the one we have performed.
- In the material and methods section it is described in detail. It reflects exactly how the sample was collected, the inclusion and exclusion criteria, the variables and the software used. All this can be considered as a strength of the present manuscript. However, as a signal and weakness, we can highlight that different important ethical considerations are not reflected. It would be necessary to indicate if the participation of the patients was voluntary or if they were offered a document with the information and informed consent. Furthermore, it would be important to point out whether the sample is representative.
Information regarding informed consent /assent has been included. All participants were informed of the study details and they consented to participate. For safety reasons, in this pilot trial the inclusion criteria were strict, and only patients with >350 CD4 were included. Results would be applicable to well controlled, ART treated patients under viral suppression, but not to untreated or immunosuppressed patients, and we have included this among other limitations in the discussion section.
The results are presented in a clear and orderly manner. Also, an interpretation of them is reflected. However, Figure 1 (line 187) and Figure 2 (line 199) display poorly. Perhaps you could improve this point.
We regret the poorly display. The figures have been uploaded separately to improve definition.
A section that gives the article quality, being a strength of it, is the discussion. In the manuscript, the results are discussed in an orderly manner, with necessary citations and rigor.
We thank you for the positive input.
In relation to the conclusions, they are clear and precise. However, we recommend indicating more precisely the implication in clinical practice of the same. Also, it would be interesting to state the lines of the future that the authors consider.
The conclusions have been extended to include future implications.

Reviewer 3 Report
In the manuscript titled "Targeting The Gut Microbiota of Vertically HIV-infected Children to Decrease Inflammation and Immunoactivation: a Pilot Clinical Trial " by Talia Sainz and colleagues, they have reported that In this exploratory study, a four-week nutritional supplementation had no significant effects in terms of decreasing inflammation, microbial translocation or T-cell activation in HIV-infected children. However, correlations were found to support the interaction between gut microbiota and the immune system. I have a few comments regarding the present manuscript.
-Please follow the author guidelines about the references
-Check the use of abbreviations, for example, ART in line 54 and later his explanation in line 59.
-The administration of probiotics or synbiotics should be implemented in the introduction, ¿how many of these studies were performed.?
-About the microbiota analyses, detailed information is required, for example, which kit was used for the extraction, how the samples were later amplified, and also why the authors have used OTUs instead ASVs
-Figure 1 resolution should be improved
-Why the authors do not show the relative abundances of the different microbial taxons
-If the authors add a treatment, the main idea is testing the efficacy, or changes that treatment produces, this is missing in the present study.
- Microbiota analyses are required in the present manuscript, measures of PcoA or alpha o beta diversity are required
-The authors have to improve special sections in their manuscript, in my opinion, the introduction lacks information regarding synbiotic formulation in HIV patients, especially in children.
Later, more detailed information on how the different processes were done. The results, explain the changes in the patients' treatment, relative abundances, microbial analyses, and finally correlations with clinical outcomes.
Discussion, explain the results and emphasize the novelty of the present study.
Author Response
-Please follow the author guidelines about the references
The references have been double-checked and adjusted to the requirements. Thank you
-Check the use of abbreviations, for example, ART in line 54 and later his explanation in line 59.
We appreciate the remark. The manuscript has been revised and all abbreviations are explained both in the abstract and the main text.
-The administration of probiotics or synbiotics should be implemented in the introduction, ¿how many of these studies were performed.?
We have included more information regarding previous data on symbiotics, although there are no specific studies in children living with HIV, which in fact emphasizes the relevance of this study. Thank you.
-About the microbiota analyses, detailed information is required, for example, which kit was used for the extraction, how the samples were later amplified, and also why the authors have used OTUs instead ASVs
As mentioned in the results section, first results of this study, and specifically the ones regarding microbial composition in HIV -infected and uninfected children, as well as evolution over time according to intervention arms have been previously published. This manuscript focuses on the inflammatory changes. Main findings regarding microbiota composition have been summarized here but we understand that results cannot be included in full, as this would result in duplicate publication. We have extended the summary to include main findings, and refer to our previous paper both for methodology and results. The methodology could be included as Supplementary material if so considered. The use of OTUs for the analysis of amplicon data to generate the community tables was the standard when the microbiome analysis was performed and subsequently published.
-Figure 1 resolution should be improved
We regret the poorly display. The figures have been uploaded separately to improve the definition.
-Why the authors do not show the relative abundances of the different microbial taxons
Please see answer above
-If the authors add a treatment, the main idea is testing the efficacy, or changes that treatment produces, this is missing in the present study.
As stated in the introduction, the main aim when testing a symbiotic nutritional supplementation here was to decrease inflammation, bacterial traslocation and improve mucosal barrier in HIV-infected children on antiretroviral treatment. The nutritional supplementation failed to achieved changes at blood level, as explained in the results section. Figure 1 and 2 summarize changes according to intervention arm in inflammatory markers, microbial traslocation markers and T-cell subsets, all non-significant.
- Microbiota analyses are required in the present manuscript, measures of PcoA or alpha o beta diversity are required
Please see the answer above
-The authors have to improve special sections in their manuscript, in my opinion, the introduction lacks information regarding synbiotic formulation in HIV patients, especially in children.
We appreciate the remark. We have included more information regarding previous data on symbiotics, although there are no specific studies in children living with HIV. We have also emphasized the need for strategies addressing inflammation in people living with HIV, which underlines the relevance of this study.
Later, more detailed information on how the different processes were done. The results, explain the changes in the patients' treatment, relative abundances, microbial analyses, and finally correlations with clinical outcomes.
Full methodology regarding the trial was published along with first results. We have tried to summarize here the trial design so that readers can understand the intervention, but have not detailed randomization process or other details regarding nutritional supplementation implementation. We have referenced our previous works both for methodology regarding microbiota analysis and for previous trial results. As mentioned before, we are open to include supplementary methods, according to the journal preferences.
The aim of this study did not include any clinical outcome, apart from a potential increase in CD4 counts (not achieved as shown in Figure 2). As al patients were and remained on treatment and the nutritional supplementation was short, we would not expect any correlation with clinical outcomes. The impact of this intervention would be mediated by a decrease in inflammation, but in the long-term in any case.
Discussion, explain the results and emphasize the novelty of the present study.
We have enlarged our interpretation of the result, emphasizing the lack of previous data and strategies to target inflammation in HIV-infected children, which underlines the relevance of this study.

Round 2
Reviewer 3 Report
Thanks to the authors for taking into account my previous comments. My main concerns this time is the resolution of Figures 1 and 2, they are difficult to follow.